# Genome-Wide Identification and Expression Profiling Analysis of the CCT Gene Family in *Solanum lycopersicum* and *Solanum melongena*

**DOI:** 10.3390/genes15111385

**Published:** 2024-10-28

**Authors:** Liangyu Cai, Rui Xiang, Yaqin Jiang, Weiliu Li, Qihong Yang, Guiyun Gan, Wenjia Li, Chuying Yu, Yikui Wang

**Affiliations:** Institute of Vegetable Research, Guangxi Academy of Agricultural Sciences, Nanning 530007, China; liangyuc5581@163.com (L.C.); xiangrui771@gmail.com (R.X.); jiangyaqin@gxaas.net (Y.J.); liweiliu@gxaas.net (W.L.); yangqihong@gxaas.net (Q.Y.); ggyun19@163.com (G.G.); 13907714662@163.com (W.L.); yuchuying@gxaas.net (C.Y.)

**Keywords:** eggplant, tomato, expression analysis, CCT, genome-wide analysis

## Abstract

CCT family genes play crucial roles in photoperiodic flowering and environmental stress response; however, there are limited reports in *Solanum* species with considerable edible and medicinal value. In this study, we conducted genome-wide characterization and expression profiling analysis of the CCT gene family in two *Solanum* species: tomato (*Solanum lycopersicum* L.) and eggplant (*Solanum melongena* L.). A total of 27 *SlCCT* and 29 *SmCCT* genes were identified in the tomato and eggplant genomes, respectively. Phylogenetic analysis showed that the CCT gene family could be divided into six subgroups (COL I, COL II, COL III, PRR, CMF I, and CMF II) in *Oryza sativa* and *Arabidopsis thaliana*. The similarity in the distribution of exon–intron structures and conserved motifs within the same subgroup indicated the conservation of *SlCCT* and *SmCCT* genes during evolution. Intraspecies collinearity analysis revealed that six pairs of *SlCCT* genes and seven pairs of *SmCCT* genes showed collinear relationships, suggesting that segmental duplication played a vital role in the expansion of the *SlCCT* and *SmCCT* family genes. *Cis*-acting element prediction indicated that *SlCCT* and *SmCCT* were likely to be involved in multiple responses stimulated by light, phytohormones, and abiotic stress. RT-qPCR analysis revealed that *SmCCT15, SlCCT6/SlCCT14*, and *SlCCT23/SmCCT9* responded significantly to salt, drought, and cold stress, respectively. Our comprehensive analysis of the CCT gene family in tomato and eggplant provides a basis for further studies on its molecular role in regulating flowering and resistance to abiotic stress, and provides valuable candidate gene resources for tomato and eggplant molecular breeding.

## 1. Introduction

Tomato (*Solanum lycopersicum* L.) and eggplant (*Solanum melongena* L.) are two typical species and have high economic importance among the *Solanum* genus [1]. They are extensively cultivated worldwide as edible vegetables. They are rich in nutrients, have pharmacological effects, and contribute significantly to human health [2,3,4]. Tomato is a common crop native to Central and South America. It is favored by many consumers for its delicious taste and high nutritional value. It also serves as a model species for researchers studying various biological processes [5]. Eggplant is the third most important herbaceous crop in Asia, with its production steadily increasing [6], and its edible parts contain vitamins, polyphenolic compounds, antioxidants, and have medicinal effects [7]. Fruits constitute the primary harvested component of tomato and eggplants, and their quality and yield are influenced by various factors. The flowering period, which serves as a critical developmental stage in the life cycle, is not only an essential part of the reproductive process but also significantly affects ecological adaptability, yield, and quality. Different stress conditions exert substantial effects, prompting plants to modify their flowering times as an adaptive strategy [8]. CCT is a key transcription factor that plays an important role in circadian responses, flowering time regulation, vernalization, and abiotic stress responses (drought, cold, and salt) [9,10,11,12,13,14,15].

The CCT gene family encompasses a nuclear localization sequence encoding 43–45 amino acids at the C-terminus, which aligns with the physiological role of transcription factors in the nucleus. Previous studies have highlighted the significance of the CCT domain in transcriptional regulation [16,17]. The CCT domain was initially identified in CONSTANS (CO) and subsequently found in TIMING OF CAB1 (TOC1) and CO-LIKE, leading to their collective designation as CCT (CO, CO-LIKE, and TOC1) [18]. According to the different conserved domains, the CCT gene family can be divided into three subfamilies: CO-LIKE (COL, contains the CCT domain and one or two zinc finger B-box [zf_B-box] domains), CCT MOTIF FAMILY (CMF, contains only the CCT domain), and PSEUDO-RESPONSE REGULATOR (PRR, contains the CCT and Response_reg domains) [10]. The zf_B-box domain, typically comprising 40 amino acid residues, plays a crucial role in protein interactions and transcriptional regulation [19]. The Response_reg domain, approximately 120 amino acids long, resembles the receiver domain of two-component response regulators. However, it lacks a distinct phosphate-accepting aspartate site in its receiver domain [20]. Transcriptional repression activity depends on the presence of functional CCT domains for target gene recognition [21]. Based on these domains, CCT family genes play a pivotal role in various aspects of growth and development.

CCT family genes have been extensively studied and their roles in the circadian clock, photoperiodic flowering, and stress responses have been elucidated [10,18]. *CO* is the first CCT family gene identified in *Arabidopsis* and is recognized for its pivotal role in regulating flowering time and circadian rhythms [10]. Functional determination of CCT family genes is intricately linked to day length (photoperiod). For instance, *HEADING DATE 1* (*HD1*) (homologous to CO) in rice has been cloned, demonstrating that long-day conditions can delay rice heading date and flowering, whereas short-day treatment can promote these processes [22]. Cloning of *Days to Heading on Chromosome 2* (belonging to the COL subfamily) has been shown to promote early flowering in rice by upregulating the expression of the florigen genes *Hd3a* and *Rice Flowering Locus T1* under long-day conditions [23]. *Ghd7* (belonging to the CMF family) acts upstream of *Early HD1* (*Ehd1*) and *Hd3a*, thereby inhibiting the expressions of *Ehd1* and *Hd3a* under long-day conditions, further affecting the reproductive development and yield of rice [24]. The PRR subfamily, which is a major component of the core oscillator of the circadian clock, plays an important role in plant flowering [20,25]. Previous studies have demonstrated that *PRR*s exhibit a transcriptional rhythm throughout the day and that their expression levels vary over time. Specifically, *AtPRR9* reaches its peak before dawn, followed by *AtPRR7*, *AtPRR5*, *AtPRR3*, and finally *AtTOC1* (as *AtPRR1*) peaking before dusk [26]. Recently, *CmNRRa* (homologous to *TOC1*) was reported to inhibit flowering in chrysanthemum [27]. Furthermore, CCT family genes play crucial roles in biological processes, such as vernalization [28,29], development of reproductive cones [30], and anthocyanin synthesis [31].

CCT has been extensively reported in plant development, such as flowering, and its potential role in stress has gradually become clear, including a pivotal role in salt [13,32], drought [11,33], cold stress [34,35,36], and the regulation of disease resistance responses through hormonal signals such as salicylic acid and ethylene signaling [37]. In addition to extensive studies on the dicotyledonous model plant *Arabidopsis thaliana*, the CCT gene family has been primarily focused in related research in various crops (rice, sorghum, maize, etc.) [10,18,38]; however, there are limited reports on horticultural crops, especially the *Solanum* genus.

In this study, we comprehensively and meticulously characterized and analyzed the CCT gene family of tomato and eggplant. This involved the identification of CCT members, their classification based on phylogenetic relationships, evolutionary analysis, tissue expression profile analysis, and examination of their responses to cold stress. These findings provide new insights into the functions of candidate genes involved in plant growth, development, and stress physiology. These findings are expected to provide a valuable foundation for future research in these areas.

## 2. Materials and Methods

### 2.1. Plant Materials and Stress Treatments

The test materials used in this study were planted in the Germplasm Resource Nursery of the Academy of Agricultural Sciences in Guangxi Zhuang Autonomous Region. For hormone treatment, five-week-old eggplant ‘177’ and tomato ‘Ailsa Craig’ (AC) (they were grown in a greenhouse under 16 h light/8 h dark conditions at 24 °C) plants were sprayed with solutions of 100 µM abscisic acid (ABA), 200 µM ethephon (ET), 100 µM 2,4-epibrassinolide (EBR), 100 µM indoleacetic acid (IAA), and 100 µM salicylic acid (SA) on their leaves, while pure water was used as a control. Leaf samples were randomly collected 1, 3, 6, 12, and 24 h after hormone treatment.

Five-week-old eggplant ‘177’ and tomato ‘AC’ plants were subjected to different abiotic stress treatments: salt (the soil was thoroughly irrigated with a solution containing 200 mM NaCl), drought (cessation of irrigation), oxidation (the young leaves were sprayed with 10 μM methyl viologen (MV) until fully saturated), cold (placement in a 4 °C growth chamber), and heat stress (placement in a 45 °C growth chamber). Leaf samples from plants subjected to abiotic stress were collected after 1, 3, 6, 12, and 24 h of treatment.

For different treatment methods, a completely randomized design was used to establish three biological replicates for each sample. Samples were collected and immediately frozen in liquid nitrogen and stored at −80 °C until subsequent experiments.

### 2.2. Genome Sequence Collection and CCT Family Gene Identification

Genome sequence files for the identification of tomato (*S. lycopersicum* L.) (version SL4.0, annotation ITAG4.0) and eggplant (*S. melongena* L.) (*S. melongena* HQ-1315 genome) were obtained from the Solanaceae Genomics Network (https://solgenomics.net/, accessed on 10 October 2023). The *Arabidopsis* CCT protein sequences retrieved from TAIR (https://www.arabidopsis.org/, accessed on 10 October 2023) were used as query sequences against the complete protein sequences of tomato and eggplant using the local BLASTP tool of the SPDE with default parameters [39]. Simultaneously, the hidden Markov model (HMM) file of the CCT protein domain (PF06203) was obtained from InterPro (https://www.ebi.ac.uk/interpro/, accessed on 13 October 2023). Tomato and eggplant CCT family members were identified from the genome database using HMMER3 (http://hmmer.janelia.org/, accessed on 13 October 2023). The results obtained from the HMM and local BLASTP were combined. Subsequently, InterPro and SMART [40] tools were employed to determine whether each candidate CCT protein had conserved domains, as described by Chen et al. [41]. Candidate proteins with partial domains were excluded from the final results. The molecular weights and isoelectric points of the CCT protein sequences were analyzed using SPDE.

### 2.3. Multi Sequence Alignments and Phylogenetic Analysis of the CCT Proteins

The full amino acid sequences of CCT from *A. thaliana* and rice (*Oryza sativa* L.) were acquired from the TAIR database (http://www.arabidopsis.org/, accessed on 26 October 2023) and Rice Genome Annotation Project (http://rice.uga.edu/, accessed on 26 October 2023), respectively, based on a previous publication [41]. The CCT protein sequences of tomato, eggplant, *Arabidopsis*, and rice were aligned using the ClustalW MEGA10 program with default settings, and phylogenetic trees were constructed using the neighbor-joining method implemented in MEGA 10 software [42] with p-distance and 1000 bootstrap replicates.

### 2.4. Gene Structure, Conserved Motifs Analysis, and Cis-Acting Regulatory Element Prediction

The full-length gene and coding sequences of the identified *CCT* genes were extracted from the genome files of tomato and eggplant using SPDE software (version 2.0). Sequence information was obtained and utilized for gene exon–intron structure analysis through the Gene Structure Display Server 2.0 online website (https://gsds.gao-lab.org/, accessed on 23 October 2023). For conserved motif analysis, MEME software (version 5.5.7) [43] was employed with a specified motif number of 20 and default settings for the others. To predict *cis*-acting regulatory elements, the promoter sequences (2 kb upstream) of *CCT* genes were examined using the PlantCARE database [44]. The bubble chart was generated by ChiPlot online software (https://www.chiplot.online/, accessed on 27 October 2023).

### 2.5. Homologous and Selection Pressure Analysis of CCT Genes

Interspecies synteny and intraspecies collinearity analyses were conducted using MCScanX in TBtools (version 2.0.0) [45], and tandem duplication was defined as described by Holub [46]. The synonymous (Ks), non-synonymous (Ka), and Ka/Ks substitution rates for each paralogous gene pair were calculated using the KaKs_Calculator 3.0 [47].

### 2.6. Expression Analysis of CCT Genes

The RNA sequencing data (leaf, stem, root, flower, and fruit of tomato and eggplant) were obtained from the European Nucleotide Archive (ENA) (https://www.ebi.ac.uk/ena/browser/home, accessed on 22 December 2023), based on published studies (PRJNA600385, PRJNA589155, PRJNA589155, PRJNA484882, PRJNA603594, PRJNA865018, PRJNA865018, PRJNA728497, PRJNA613773, and PRJNA477924), for the analysis of tissue expression of CCT genes. All transcriptome data were mapped to the tomato and eggplant reference genomes (ITAG4.0 and *S. melongena*-HQ) using HISAT2 [48]. Expression data (transcripts per kilobase of exon model per million mapped reads, TPM) were normalized to log2 (TPM + 1), and heat maps were generated using TBtools. The detailed processing workflow is described by Xiang et al. [49].

### 2.7. RNA Isolation and Real-Time Quantitative PCR (RT-qPCR) Analysis

RNA was extracted using the OmniPlant RNA Kit (CWBIO, Beijing, China). The RNA OD value was measured using a NanoDrop spectrophotometer (Thermo Fisher Scientific, Waltham, MA, USA), and the quality was evaluated via agarose gel electrophoresis. Genomic DNA (gDNA) was removed using PrimeScript RTase (Trans Gen Biotech, Beijing, China), and cDNA synthesis was performed using the UEIris II RT-PCR System for First-Strand cDNA Synthesis Kit (US Everbright^®^ Inc., Suzhou, China). Specific primers for RT-qPCR were designed using Primer3web (https://primer3.ut.ee/, version 4.1.0, accessed on 7 December 2023), and their sequences are provided in Appendix A (the sequences of *CCT1* and *CTT2* genes of tomato and eggplant are so similar that the same primer pair was used). *Actin* served as an internal reference gene. Each RT-qPCR reaction was conducted in a total volume of 20 μL, including 1 μL of each primer, 1 μL of cDNA template, 10 μL of 2× Fast Super EvaGreen^®^ qPCR Mastermix (US Everbright Inc., Suzhou, China), and 7 μL of ddH_2_O. Each sample included three biological and three technical replicates. RT-qPCR was performed on a LightCycler^®^96 (Roche, Shanghai, China) using the following amplification parameters: 95 °C for 120 s, followed by 45 cycles of 95 °C for 5 s and 59 °C for 30 s. Relative gene expression levels were calculated using the 2^−ΔΔCT^ method [50]. Heat maps were generated using TBtools.

## 3. Results

### 3.1. Basic Information of CCT Family Members of Tomato and Eggplant

Analysis of the genome information of *S. lycopersicum* and *S. melongena* obtained from the Solanaceae Genomics Network database revealed 27 and 29 putative CCT members, respectively, which were renamed based on the order of chromosomal location (Table 1). The *SlCCT1*–*SlCCT27* gene coding sequence length was between 801 and 2043 bp, and the *SmCCT1*–*SmCCT29* gene coding sequence length was between 366 and 2346 bp. The size of a protein is generally proportional to the length of its amino acid sequence. The molecular weight of the 27 *SlCCT* and 29 *SmCCT* genes varied from 30,693.37 to 74,560.97 Da and 13,689.47 Da to 85,395.89 Da, respectively. The number of CCT genes across different subfamilies is comparable between tomato and eggplant, with 8 CMF subfamily genes in tomato and 9 in eggplant, 12 COL subfamily genes in tomato and 13 in eggplant, and 5 PRR subfamily genes in tomato and 6 in eggplant. More detailed information regarding the gene locus ID, chromosomal position, isoelectric points, and subfamily classification is shown in Table 1.

### 3.2. Phylogenetics, Gene Structure Analysis, and Conserved Motif Identification of the CCT Family

To understand the evolutionary relationships between tomato, eggplant, rice, and *Arabidopsis*, a maximum likelihood phylogenetic tree was constructed based on CCT acid sequences (Figure 1). It can be clearly clustered into six subgroups: COL I, COL II, COL III, PRR, CMF I, and CMF II, and most members of the same subfamily were clustered into the same subgroups. In particular, OsCMF2, OsCMF8, OsCMF11, AtCFM2, and AtCMF5 were not divided into subgroups because of their low bootstrap values. As expected, tomato and eggplant had higher CCT gene homology than the monocot plant rice. In addition, in the same subgroups, the number of CCT genes in tomato and eggplant was the same but different from that in *Arabidopsis*, such as the COL I (5), COL II (3), CMF I (2), and CMF II (6) subgroups, indicating that the four subgroups experienced the same changes in the evolutionary pathways.

The exon–intron and conserved motif distribution patterns revealed the conservation and diversity of CCT family genes. A singular phylogenetic tree, exon–intron distribution patterns, and motif distribution patterns of CCT family genes of tomato and eggplants are shown in Figure 2. The exon numbers of the tomato and eggplant CCT genes were 1–8 and 2–8, respectively. Two of the COL II subgroup members of eggplant and tomato had only two exons, indicating that they were conserved during evolution. The number of exons in the other subgroup members was different. Notably, the two pairs of genes of the CMF I subgroup members, *SmCCT3*/*SlCCT3* and *SmCCT16*/*SlCCT16*, had the same number of exons and were located at the same number on their respective chromosomes (Table 1, Figure 2), indicating that the evolutionary pathway of these two pairs of genes was very similar.

The type and distribution of conserved motifs often offer valuable insights into potential gene functions (Figure 2). In the motif distribution patterns of CCT proteins in tomato and eggplant, both contained the typical conserved domain of CCT proteins, the CCT domain (both motifs 1), and both were located at the C-terminus. In addition to the CCT domain, the COL subfamily had a zf_B-box domain (motifs 2 and 4 in tomato, motif 2 in eggplant) located at the N-terminus. Similarly, the typical Response_reg domain of the PRR subfamily was located at the N terminus. Overall, the motif distribution patterns in different subgroups were conserved.

### 3.3. Chromosomal Location and Expansion Patterns of the CCT Family

*CCT* genes were mapped onto tomato (Slchr) and eggplant (E) chromosomes using genome annotation information. The distribution of *CCT* genes on the chromosomes of tomato and eggplants was very similar. In both the species, chromosome 1 did not contain any *CCT* genes, and the rest of the chromosomes were unevenly distributed with *CCT* genes. The same number of *CCT* genes was found on the same chromosome location in tomato and eggplants, such as Slchr02/E02, Slchr05/E05, Slchr06/E06, Slchr07/E07, and Slchr08/E08 (Figure 3). Furthermore, five pairs of segmental duplication genes (*SlCCT6/14*, *SlCCT7/8*, *SlCCT17/25*, *SlCCT20/26*, and *SlCCT22/27*) and one pair of tandem duplication genes (*SICCT1/2*) were identified in the tomato genome (Tabel S1), whereas six pairs of segmental duplication genes (*SmCCT7/14*, *SmCCT10/25*, *SmCCT12/19*, *SmCCT13/23*, *SmCCT21/Smechr1002246.1*, and *SmCCT24/27*) and one pair of tandem duplication genes (*SmCCT1/2*) were identified in the eggplant genome (Appendix A), indicating that segmental duplication plays a major role in the expansion of the *CCT* gene family in both tomato and eggplant. Moreover, all the duplication events belonged to the same subgroup, except for *SlCCT20/26*. The Ka/Ks ratio results revealed (Appendix A) that all gene pairs had Ka/Ks ratios < 1 (0.06–0.87), with the exception of *SmCCT10/25* (Ka/Ks = 2.06), indicating that purifying selection had acted upon these genes.

### 3.4. Analysis of Evolutionary Relationships Within Arabidopsis, Tomato, and Eggplant

To obtain more information on the evolution of *CCT* genes, a comparative synteny map of *Arabidopsis*, tomato, and eggplant was constructed. As shown in Figure 4, tomato and eggplant *CCT* genes shared 25 (Figure 4A, Appendix A) and 27 (Figure 4A, Appendix A) synteny gene pairs with *Arabidopsis*, respectively. Additionally, 35 synteny gene pairs were identified between tomato and eggplants (Figure 4B, Appendix A). Interestingly, seven *SlCCT* and six *SmCCT* genes were paired with more than one *Arabidopsis* homolog (Appendix A), and nine *SmCCT* genes were paired with more than one tomato homolog (Appendix A). It is speculated that these genes originated from a common ancestral gene in evolutionary history and may have retained similar functions.

### 3.5. Cis-Acting Element Analysis of the CCT Family

*Cis*-elements in the promoter region may provide insights into a gene’s potential function and regulatory roles. As shown in Figure 5, the 18 *cis*-acting elements in the promoter region of the CCT family can be divided into four categories: hormonal regulation, plant development, stress regulation, and light regulation. In both tomato and eggplant, there were significantly more ABA-responsive element numbers (ABRE) and light-responsive element numbers (G-box) within the CCT family genes compared with other types of elements (Figure 5, Appendix A), indicating that CCT family genes may play potential roles in the ABA signaling pathway and adaptation to environmental light. Next was the methyl jasmonate response element (CGTCA and TGACG motifs). In particular, SlCCT11 contained the most (7) TCA elements (salicylic acid response). In stress regulation, the number of anaerobic-responsive elements was the highest, while both drought-responsive elements (MBS) and low-temperature-responsive elements (LTR) were also predicted in *SmCCT* and *SlCCT*. Furthermore, the distribution of plant development-related elements indicated the potential role of *SmCCT* and *SlCCT* in tomato and eggplant growth and development, respectively.

### 3.6. Expression Profiles of CCT Family Genes in Different Tissues

Transcriptomic analysis of CCT family genes in fruits, roots, flowers, leaves, and stems were performed to provide insights into their potential roles in growth and development (Figure 6). The results showed that *SlCCT* and *SmCCT* were highly expressed in various tissues; *SlCCT9*, *SlCCT11*, *SlCCT12*, *SlCCT20*, *SlCCT21*, *SlCCT26*, *SmCCT5*, *SmCCT9*, *SmCCT12*, *SmCCT15*, *SmCCT19*, *SmCCT22*, and *SmCCT26* were expressed at high levels in the five tissues. Notably, *SlCCT9* and *SlCCT26* demonstrated particularly elevated expression levels in flowers and leaves, respectively; *SmCCT12* demonstrated particularly elevated expression levels in fruits and leaves. Some genes displayed distinct tissue-specific expression patterns; for instance, *SlCCT22*, *SlCCT27*, and *SmCCT28* were highly expressed in flowers.

### 3.7. Expression Analysis of CCT Family Genes Under Hormone Treatment

Phytohormones are present in minute concentrations and have a significant influence on plant growth, development, reproduction, and response to environmental stressors. Subtle alterations in phytohormone levels frequently manifest as physiological or phenotypic variations in plants. To deepen our understanding of the mechanism of action of the CCT gene family during growth and development, its expression patterns were analyzed under varying hormonal conditions (ABA, EBR, ET, IAA, and SA) using RT-qPCR. In total, 18 *SmCCT* and 19 *SlCCT* genes were randomly selected for gene expression analysis. As shown in Figure 7, most *CCT* genes in eggplant and tomato responded to exogenous hormone treatments. Notably, compared with *SlCCT* genes, the majority of *SmCCT* genes in eggplant were strongly induced by these treatments. For example, *SmCCT15* and *SmCCT21* were upregulated within 24 h of hormone application. Interestingly, *SmCCT4*, *SmCCT24*, and *SmCCT28* displayed a similar response pattern across different hormone treatments, characterized by a wave-like expression pattern: initial downregulation, followed by upregulation and subsequent downregulation, with strong expression levels observed 12 h post-treatment.

In tomato, *SlCCT* genes were particularly responsive to ABA and salicylic acid compared with other hormones. For example, the expression of *SlCCT14* was suppressed by EBR, ET, and IAA but was induced by ABA. Meanwhile, *SlCCT5* was upregulated by most hormone treatments, whereas *SlCCT7*, *SlCCT16*, and *SlCCT17* were generally inhibited by various hormone applications.

### 3.8. Expression Analysis of CCT Family Genes Under Abiotic Stress Treatment

To preliminarily investigate the involvement of the CCT gene family in various abiotic stresses, we analyzed the expression levels of CCT family genes under cold, MV, drought, heat, and salt stress conditions across different treatment periods using RT-qPCR (Figure 8). Generally, most *CCT* genes respond to stress treatments, either through induction or inhibition. In eggplants, most *SmCCT* genes were induced and upregulated. For example, *SmCCT12* and *SmCCT15* were upregulated under various stress conditions. However, some genes displayed differential responses depending on the treatment. For example, the expressions of *SmCCT16* and *SmCCT27* were inhibited by cold and drought stress but upregulated in response to heat and salt stress. Notably, *SmCCT24* and *SmCCT28* showed distinct expression patterns, with downregulation in the early stages of MV, drought, heat, and salt treatments, followed by substantial upregulation in the later stages, indicating a potential feedback regulation mechanism. The inhibition of *SmCCT27* under cold, MV, and drought stress conditions was remarkable, particularly under drought conditions.

In tomato, the expression of most *SlCCT* genes was inhibited by salt stress, particularly in comparison with *SmCCT* genes. Only a few genes, *SlCCT5*, *SlCCT14*, and *SlCCT21*, were upregulated during the later stages of salt stress. However, under drought and heat stress conditions, most *SlCCT* genes were induced, including *SlCCT5*, *SlCCT6*, *SlCCT11*, *SlCCT12*, and *SlCCT14*.

## 4. Discussion

CCT family genes have been primarily researched in model plants such as *A. thaliana* and crop plants, demonstrating a key role in the regulation of photoperiodic flowering, product yield, and stress response [10,18,51]. With the continuous advancement and affordable pricing of next-generation sequencing technology, genome data from various species are now well-assembled and annotated. The CCT gene family has been characterized in numerous plants, including wheat (*Triticum aestivum* L.) [52], rice [53], *Raphanus sativus* [54], poplar [41], Pigeonpea (*Cajanus cajan* (L.) Millsp) [55], alfalfa (*Medicago truncatula* L.) [56], and foxtail millet (*Setaria italica* L.) [57]. However, a comprehensive genome-wide identification and characterization of the genus *Solanum*, especially in tomato and eggplants, has not been performed. In this study, we identified 27 *SlCCT* and 29 *SmCCT* full-length CCT coding sequences in the tomato and eggplant genomes, respectively (Table 1), renamed them based on chromosomal location, and further analyzed their evolutionary relationships, gene structures, conserved motifs, chromosome locations, *cis*-acting elements, interspecies synteny, intraspecies collinearity, tissue-specific expression, and expression patterns under cold stress conditions.

To understand evolutionary relationships, a phylogenetic tree was constructed for tomato, eggplants, *Arabidopsis*, and rice. Phylogenetic analysis showed that most CCT family members in rice branched independently, indicating that the CCT genes of tomato, eggplant, and *Arabidopsis* have higher homology and a more conserved evolutionary process. Further analysis revealed that the CMF, COL, and PRR subfamilies were divided into two (CMF I and CMF II), three (COL I, COL II, and COL III), and one subgroup, respectively, in both tomato and eggplant. This clustering trend has also been observed in *R. sativus* [54] and pear (*Pyrus bretschneideri* Rehd.) [58], indicating that our results are reliable. In addition, the PRR groups exhibited conserved motif distribution patterns and exon–intron structures, suggesting that the PRR subfamily genes were more conserved in both evolution and function. Previous comprehensive comparisons and phylogenetic analyses of the CMF, COL, and PRR families in *Poaceae* have indicated that COL and CMF have undergone more evolutionary changes [59]. This finding further confirms the evolutionary and functional diversity within the CMF and COL subfamilies of tomato and eggplant.

Gene duplication events are common biological events crucial for the expansion of gene families. They serve as critical mechanisms for neofunctionalization and functional divergence during evolution [60,61]. In this study, collinearity analysis revealed five pairs of segmental duplications and one pair of tandem duplications in the tomato genome (Figure 3A, Appendix A) and six pairs of segmental duplications and one pair of tandem duplications in the eggplant genome (Figure 3B, Appendix A), suggesting that segmental duplication events were the major force for the expansion of *CCT* genes. Among them, six pairs of duplicated genes in tomato and two pairs of duplicated genes in eggplant belonged to the same subfamily. Interestingly, the remaining gene pairs did not belong to the same eggplant subfamily. For example, *SmCCT1* and *SmCCT2* belonged to the COL and CMF groups, respectively. A possible reason for this finding has been reported in previous studies. As a result of double-stranded DNA breaks, COL gene evolution continues to lead to B-box degradation, in which the B-box domain is reduced from two to one and then to zero [59], becoming a CMF subfamily gene. The rate of evolution of the *COL* gene appears to have accelerated in later stages of evolution, possibly because of frequent gene duplications [62]. Furthermore, Ka/Ks analysis indicated that except for *SmGLK10/25*, all duplicated gene pairs were subjected to purifying selection (Appendix A). This implies that gene duplication events experienced strong purifying constraints throughout evolution, highlighting the significant influence of these constraints on duplication events.

Synteny analysis provides insights into gene functions. In this study, interspecies synteny analysis revealed 25 orthologous gene pairs between *Arabidopsis* and tomato (Figure 4A, Appendix A) and 27 orthologous gene pairs between *Arabidopsis* and eggplant (Figure 4A, Appendix A). According to previous research, *ATCOL13* (*AT2G47890*), an orthologous gene of *SmCCT22* and *SlCCT21*, acts as a positive regulator of R light-mediated inhibition of hypocotyl elongation [63]. *CO* (*AT5G15840*), an ortholog of *SmCCT1*, is involved in regulating flowering time in *Arabidopsis* [51]. *COL4* (*AT5G57660*), the orthologous gene of *SmCCT15* and *SlCCT15*, plays a role in the salt stress response through the ABA-dependent signaling pathway in *Arabidopsis*. Overexpression of *COL4* enhances salt tolerance [32]. *AtPRR5/7/9*, the orthologous genes of *SmCCT6*, *SmCCT9*, and *SmCCT21*, function as transcriptional repressors of the *Arabidopsis* circadian clock [64]. These results were consistent with various *cis*-acting elements, such as G-box responsive light in *SmCCT22* and *SlCCT21* promoters and ABRE-responsive ABA in *SmCCT15* and *SlCCT15* promoters (Figure 5, Appendix A). Furthermore, 35 orthologous gene pairs were identified between tomato and eggplant (Figure 4B, Appendix A), suggesting a strong relationship between CCT family genes in tomato and eggplant. These data provide valuable information to support further research on the functions of CCT family genes.

The abundance of tissue-specific expression offers insights into the biological functions of genes [65]. RNA-seq data analysis revealed that the expression of *SlCCT* and *SmCCT* genes was ubiquitous (Figure 6). The transcriptional profiling results indicated the potential involvement of *SlCCT* and *SmCCT* genes in tomato and eggplant organ development. Interestingly, some collinear gene pairs exhibited similar expression patterns. For instance, *SlCCT22* and *SlCCT27* were exclusively expressed in flowers, whereas *SmCCT12* and *SmCCT19* were highly expressed in all the tested tissues, implying functional redundancy. A substantial body of research has demonstrated that the CMF subfamily is involved in the regulation of flowering time. Specifically, genes from this subfamily delay flowering in several plant species, including rice (*Ghd7*), maize (*ZmCCT9*, *ZmCCT10*), and sorghum (*SbGhd7*) [10]. Consequently, the collinear gene pairs *SlCCT22* and *SlCCT27*, members of the CMF subfamily, are likely to play a significant role in the development of floral organs. However, the precise functions of these genes require further validation.

In addition to their roles in growth and development, certain *CCT* genes are also involved in abiotic stress responses. *Arabidopsis AtCOL4* positively regulates salt stress response in an ABA-dependent manner [32]. In the present study, *SmCCT15*, a member of the COL I subfamily, exhibited a significant response to both salt stress (Figure 8) and ABA treatments (Figure 7). The presence of an ABA-responsive *cis*-acting element (ABRE) in *SmCCT15* (Figure 5) further supported its potential role in salt stress regulation. Similarly, within the PRR subfamily, *OsTOC1* has been reported to respond to drought stress [66], and *SlCCT6* and *SlCCT14* from the same subfamily showed significant responses to drought treatment (Figure 8). *SlCCT23*, which contains cold-responsive elements (LTR; Appendix A), and its homologous gene *SmCCT9* was upregulated following cold treatment. Furthermore, mutants of *Arabidopsis prr9prr7prr5* exhibit enhanced tolerance to cold [34], suggesting that *SlCCT23* and *SmCCT9* may play a role in cold stress resistance. Interestingly, the expression pattern of CCT family genes is time-dependent. For example, under salt stress, the expression of certain genes, such as *SmCCT24* and *SmCCT28*, peaks at 12 h, but is significantly downregulated at 1 and 3 h. This suggests that some CCT genes may undergo feedback regulation during stress responses. Additionally, different members of the *SmCCT* and *SlCCT* family genes exhibit distinct response patterns under the same stress conditions. For instance, under cold stress, *SmCCT23* is significantly upregulated after 6 h, whereas *SmCCT8* is downregulated, indicating possible functional differentiation among gene family members. In particular, this study is the first to comprehensively report the expression patterns of CCT family genes under oxidative stress. Notably, the expression pattern in eggplant is similar to that observed under salt stress, while in tomato, distinct differences were observed between oxidative and salt stress responses. These findings suggest a functional divergence of CCT family genes between eggplant and tomato. Taken together, these genes exhibited varying degrees of response to other treatments, highlighting the functional diversity of *CCT* genes. The observed expression patterns further underscore the significant roles of *CCT* genes in abiotic stress responses.

## 5. Conclusions

In this study, the CCT gene family in Solanaceae plants, including tomato and eggplants, was analyzed, and 27 *SlCCT* and 29 *SmCCT* genes were identified, respectively. These genes were categorized into five subfamilies based on their phylogenetic relationship with CCT proteins in *Arabidopsis* and rice. Collinearity analysis and phylogenetic relationships of *CCT* genes provided valuable insights into the evolutionary characteristics of *SlCCT* and *SmCCT* genes. The expression patterns of *SlCCT* and *SmCCT* genes across different tissues and organs, as well as in response to hormone treatments and abiotic stress conditions, offer a deeper understanding of their potential roles in plant growth and development. This research will facilitate future functional studies on *SlCCT*/*SmCCT* genes and serve as a reference for molecular breeding aimed at enhancing the environmental adaptability and stress resistance of tomato and eggplant species.

## Figures and Tables

**Figure 1 genes-15-01385-f001:**
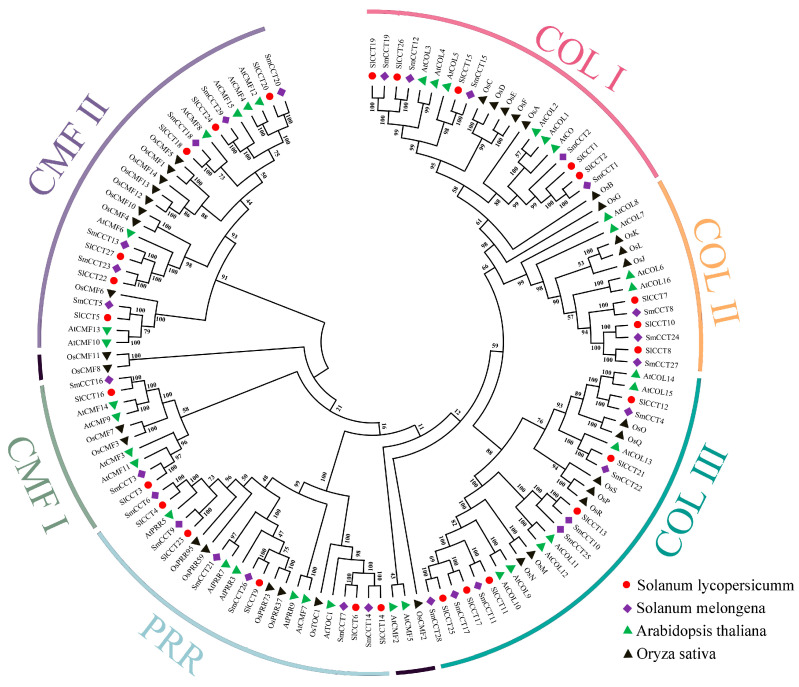
Phylogenetic analysis of CCT proteins from eggplant and tomato. Circles, diamonds, green triangle, and black triangle represent *S. lycopersicum*, *S. melongena*, *Arabidopsis thaliana*, and *O. sativa* proteins, respectively. The numbers near the branches denote bootstrap values.

**Figure 2 genes-15-01385-f002:**
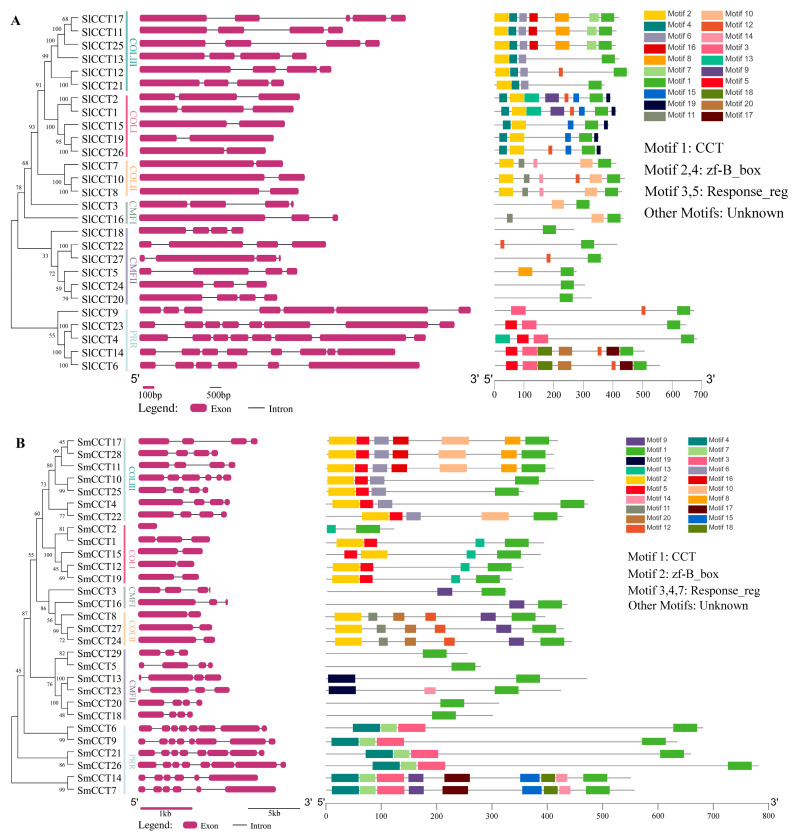
Phylogenetic trees, exon–intron distributions, and motifs analyses of *SlCCT* (**A**) and *SmCCT* (**B**). Pink boxes represent exons and black lines denote introns. Rectangles with different colors represent different motifs and motif annotation was carried out using InterPro (version 95.0).

**Figure 3 genes-15-01385-f003:**
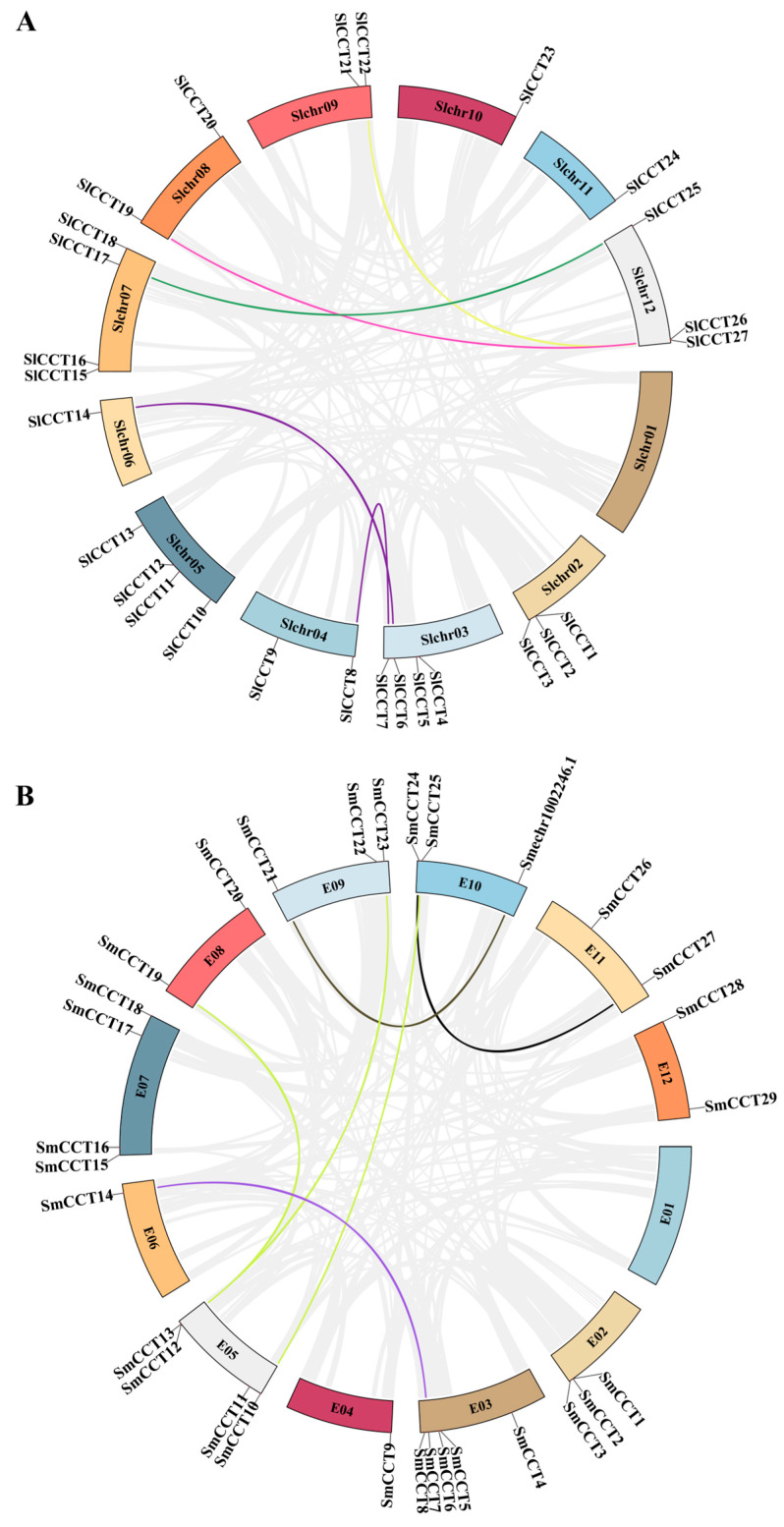
Collinearity analyses and chromosomal distributions of *SlCCT* (**A**) and *SmCCT* (**B**) genes. Colored bars connecting two chromosomal regions denote collinear regions; the corresponding genes on two chromosomes are regarded as segmental duplications. Background gray lines show the collinear blocks within *S. lycopersicum* and *S. melongena* genomes.

**Figure 4 genes-15-01385-f004:**
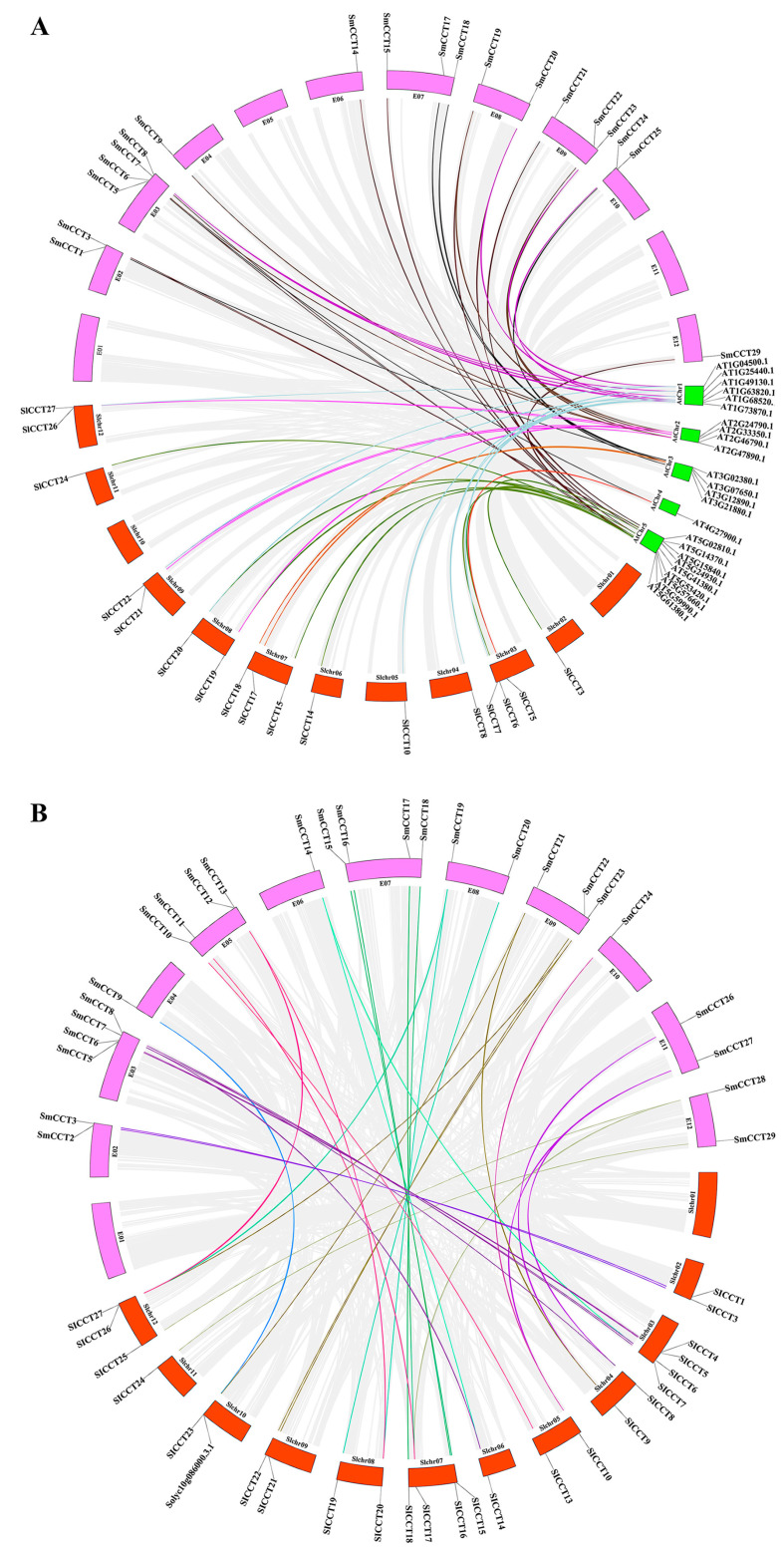
Synteny analyses of *A. thaliana*, *S. lycopersicum*, and *S. melongena*. (**A**) Synteny analysis of CCT genes in *A. thaliana* and *S. lycopersicum*/*S. melongena*. (**B**) Synteny analysis of CCT genes in *S. melongena* and *S. lycopersicum*. Colored bars connecting two chromosomal regions denote syntenic regions; background gray lines show the syntenic blocks within *A. thaliana* and *S. lycopersicum*/*S. melongena* genome and *S. lycopersicum* and *S. melongena* genome.

**Figure 5 genes-15-01385-f005:**
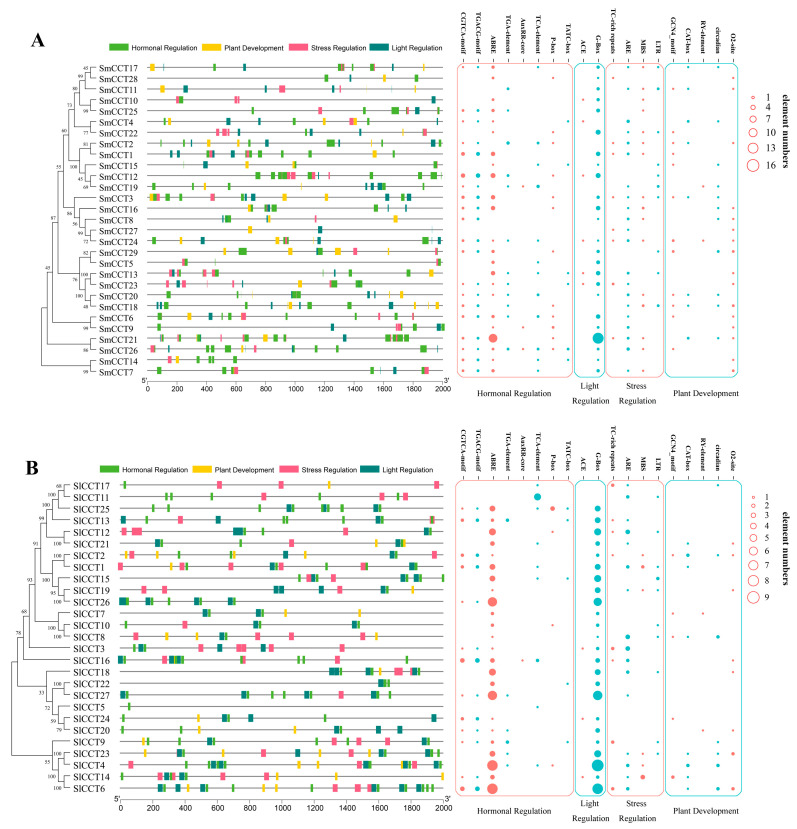
*Cis*-acting element analyses of *SlCCT* (**A**) and *SmCCT* (**B**). Different color squares represent the distributions of different *cis*-acting elements in the *CCT* gene promoter. Circles of different sizes represent the numbers of different promoters of *cis*-elements in the *CCT* genes.

**Figure 6 genes-15-01385-f006:**
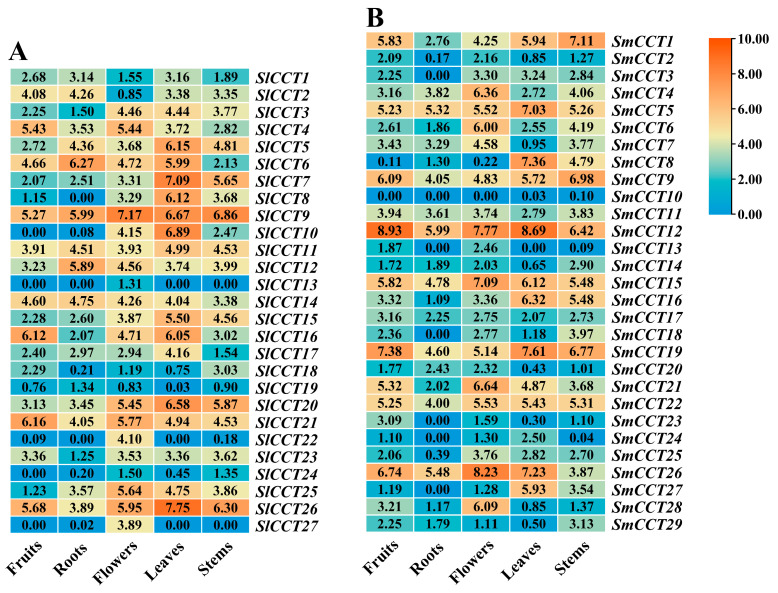
Expression profiles of *SlCCT* (**A**) and *SmCCT* (**B**) in five representative tissues (fruits, roots, flowers, leaves, and stems). The heat maps were generated based on the log2 (TPM + 1) values of *CCT* genes. Orange and blue indicate higher and lower transcript abundances, respectively.

**Figure 7 genes-15-01385-f007:**
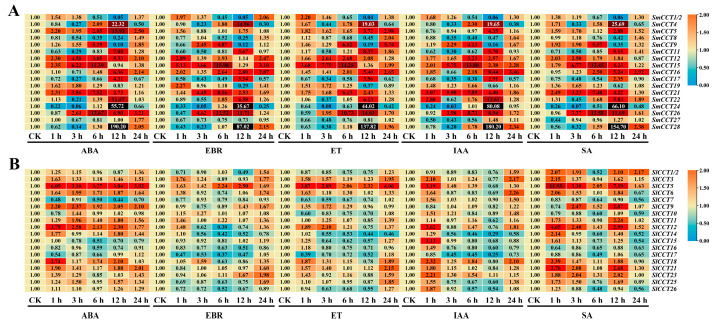
RT-qPCR expression analyses of (**A**) *SmCCT* and (**B**) *SlCCT* under hormone (ABA, EBR, ET, IAA, and SA) treatments. Leaves were collected 1, 3, 6, 12, and 24 h after treatment.

**Figure 8 genes-15-01385-f008:**
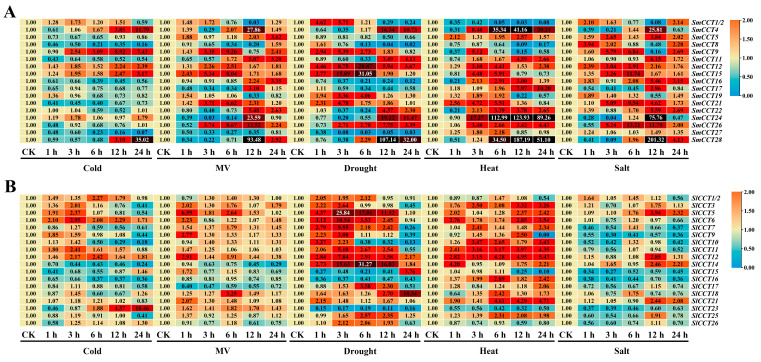
RT-qPCR expression analyses of (**A**) *SmCCT* and (**B**) *SlCCT* genes under abiotic stress (cold, MV, drought, and salt) treatments. Leaves were collected 1, 3, 6, 12, and 24 h after treatment.

**Table 1 genes-15-01385-t001:** Details of CCT genes in tomato and eggplant.

Gene Name	Gene Locus ID	Chr.	Start	End	CDS Length (bp)	ORF (aa)	pI	MW (Da)	Exon	Subfamily
Tomato genome										
*SlCCT1*	Solyc02g089520.2.1	2	49,357,153	49,358,925	1230	409	5.07	45,676.14	3	COL
*SlCCT2*	Solyc02g089540.3.1	2	49,365,916	49,368,618	1176	391	5.57	43,435.87	3	COL
*SlCCT3*	Solyc02g093590.3.1	2	52,454,166	52,457,500	963	320	6.69	36,678.42	4	CMF
*SlCCT4*	Solyc03g081240.3.1	3	46,704,175	46,709,293	2043	680	6.29	74,560.97	8	PRR
*SlCCT5*	Solyc03g083400.3.1	3	47,851,661	47,855,589	828	275	5.39	30,693.38	4	CMF
*SlCCT6*	Solyc03g115770.3.1	3	59,800,548	59,806,781	1662	553	6.24	62,852.54	6	PRR
*SlCCT7*	Solyc03g119540.3.1	3	62,565,853	62,567,474	1227	408	5.17	46,692.38	2	COL
*SlCCT8*	Solyc04g007210.3.1	4	940,495	942,888	1287	428	5.33	48,718.86	2	COL
*SlCCT9*	Solyc04g049670.4.1	4	42,073,726	42,080,420	2013	670	6.67	72,773.76	7	PRR
*SlCCT10*	Solyc05g009310.3.1	5	3,490,063	3,492,184	1314	437	5.53	49,815.17	2	COL
*SlCCT11*	Solyc05g020020.4.1	5	25,936,683	25,942,623	1242	413	5.58	44,846.46	4	COL
*SlCCT12*	Solyc05g024010.3.1	5	30,361,734	30,365,044	1359	452	6.79	49,702.34	4	COL
*SlCCT13*	Solyc05g046040.2.1	5	57,493,969	57,496,194	1260	419	4.97	46,262.16	4	COL
*SlCCT14*	Solyc06g069690.4.1	6	41,034,051	41,040,009	1515	504	5.96	56,637.70	8	PRR
*SlCCT15*	Solyc07g006630.4.1	7	1,468,325	1,470,183	1158	385	6.37	42,472.11	2	COL
*SlCCT16*	Solyc07g008540.3.1	7	3,439,593	3,443,593	1299	432	8.59	48,639.07	3	CMF
*SlCCT17*	Solyc07g045180.4.1	7	58,174,801	58,183,106	1257	418	5.41	46,133.13	5	COL
*SlCCT18*	Solyc07g066510.4.1	7	67,767,417	67,768,873	801	266	6.30	30,869.02	4	CMF
*SlCCT19*	Solyc08g081350.2.1	8	1,154,652	1,156,466	981	326	5.41	37,809.01	4	CMF
*SlCCT20*	Solyc08g006530.4.1	8	62,540,188	62,542,440	1053	350	5.28	38,771.55	2	COL
*SlCCT21*	Solyc09g074560.3.1	9	62,591,354	62,593,707	1122	373	5.80	42,226.55	4	COL
*SlCCT22*	Solyc09g090650.3.1	9	66,199,248	66,202,425	1236	411	4.97	46,665.82	4	CMF
*SlCCT23*	Solyc10g005030.4.1	10	64,712,891	64,719,450	1938	645	5.87	71,722.29	8	PRR
*SlCCT24*	Solyc11g072850.2.1	11	54,102,808	54,104,700	912	303	8.66	34,820.54	3	CMF
*SlCCT25*	Solyc12g006240.2.1	12	764,033	771,486	1227	408	5.25	45,076.73	4	COL
*SlCCT26*	Solyc12g096500.2.1	12	64,913,638	64,915,137	1077	358	5.53	39,504.47	2	COL
*SlCCT27*	Solyc12g096940.3.1	12	65,167,779	65,169,533	1095	364	4.84	41,112.77	4	CMF
Eggplant genome										
*SmCCT1*	Smechr0202811.1	2	72,045,312	72,047,478	1179	392	5.56	43,400.53	3	COL
*SmCCT2*	Smechr0202813.1	2	72,065,225	72,065,742	366	121	9.84	13,689.47	1	CMF
*SmCCT3*	Smechr0203114.1	2	74,514,378	74,517,728	969	322	7.80	37,301.34	4	CMF
*SmCCT4*	Smechr0300833.1	3	27,079,307	27,082,841	1419	472	6.52	51,793.58	4	COL
*SmCCT5*	Smechr0302191.1	3	81,810,396	81,814,608	840	279	5.13	31,180.83	4	CMF
*SmCCT6*	Smechr0302289.1	3	83,057,886	83,062,489	2046	681	6.60	75,111.82	8	PRR
*SmCCT7*	Smechr0303042.1	3	90,537,774	90,544,281	1668	555	5.86	62,868.41	6	PRR
*SmCCT8*	Smechr0303381.1	3	93,516,535	93,518,191	1191	396	5.48	45,025.45	2	COL
*SmCCT9*	Smechr0400057.1	4	597,191	603,333	1902	633	5.75	70,114.52	8	PRR
*SmCCT10*	Smechr0500021.1	5	412,819	415,930	1446	481	5.86	54,100.51	6	COL
*SmCCT11*	Smechr0500812.1	5	9,998,348	10,004,592	1236	411	5.83	45,198.99	4	COL
*SmCCT12*	Smechr0502518.1	5	78,912,341	78,913,477	1065	354	5.54	38,831.63	2	COL
*SmCCT13*	Smechr0502527.1	5	79,005,173	79,008,049	1413	470	4.91	52,659.22	4	CMF
*SmCCT14*	Smechr0602414.1	6	82,984,522	82,989,926	1650	549	6.12	62,895.45	6	PRR
*SmCCT15*	Smechr0700024.1	7	588,003	590,102	1161	386	6.10	42,475.22	2	COL
*SmCCT16*	Smechr0700295.1	7	5,761,294	5,765,503	1308	435	8.44	48,783.17	3	CMF
*SmCCT17*	Smechr0701630.1	7	90,210,322	90,218,565	1248	415	5.25	45,860.36	4	COL
*SmCCT18*	Smechr0702757.1	7	106,616,442	106,618,308	900	299	6.19	34,329.68	4	CMF
*SmCCT19*	Smechr0800344.1	8	4,423,893	4,426,265	1008	335	5.66	37,428.26	2	COL
*SmCCT20*	Smechr0802394.1	8	85,099,392	85,102,284	936	311	5.48	35,946.20	4	CMF
*SmCCT21*	Smechr0900307.1	9	4,317,509	4,324,142	1977	658	6.47	72,385.36	8	PRR
*SmCCT22*	Smechr0902033.1	9	81,226,910	81,230,720	1281	426	6.29	48,229.54	4	COL
*SmCCT23*	Smechr0902417.1	9	86,422,926	86,428,176	1272	423	4.73	47,453.64	4	CMF
*SmCCT24*	Smechr1000119.1	10	1,499,166	1,501,192	1332	443	5.42	50,489.79	2	COL
*SmCCT25*	Smechr1000256.1	10	3,115,595	3,118,657	1065	354	6.23	39,110.48	4	COL
*SmCCT26*	Smechr1101238.1	11	35,217,776	35,261,546	2346	781	6.61	85,395.89	8	PRR
*SmCCT27*	Smechr1102098.1	11	91,435,206	91,437,439	1287	428	5.55	48,757.88	2	COL
*SmCCT28*	Smechr1200014.1	12	185,762	190,529	1227	408	5.32	44,931.49	4	COL
*SmCCT29*	Smechr1201501.1	12	68,005,632	68,007,547	765	254	6.13	28,959.91	3	CMF

## Data Availability

The original contributions presented in the study are included in the article/Appendix A, further inquiries can be directed to the corresponding author.

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
