# Peer review of "Genome-Wide Identification and Expression Profiling Analysis of the CCT Gene Family in Solanum lycopersicum and Solanum melongena"

_genes, 2024, doi:10.3390/genes15111385_

Round 1

Reviewer 1 Report

Comments and Suggestions for Authors

Summary

The authors provided a comprehensive bioinformatic analysis of the CCT family genes. While the bioinformatic methodology was thoroughly detailed, there are a few issues that need to be addressed regarding the plant material and stress induction process. Additionally, the Discussion section should be briefly expanded to provide more depth. I have recommended "major revisions" to provide you more time for a careful revision of the manuscript. Once these revisions are made, the manuscript will be gladly accepted.

Abstract

Line 12: Use the plural form: "CCT family genes" to ensure consistency.

Line 12-13: Reorganize these two sentences. First, introduce the importance of the CCT family genes and then discuss their role in the Solanum genus.

Line 23: Similarly, change "CCT family gene" to "CCT family genes."

Introduction

Line 43: Use "tomato" instead of "tomatoes" when referring to the species itself.

Line 48: Start a new paragraph to introduce the CCT family. Conclude the previous paragraph by briefly mentioning that growers attempt to mitigate abiotic stress using rootstocks. You may consider incorporating references like (https://doi.org/10.3390/agronomy13112705 and https://doi.org/10.1080/01448765.2014.964317) if relevant to your work.

Line 97: Introduce in a new paragraph the aim of your study. Also, consider moving lines 101-103 to the Conclusion section, or removing them from the Introduction if they seem repetitive.

Materials and Methods

Provide more information about the plant material used, including the source (seed company or nursery) and the initial material (seeds, plantlets).

Stress induction details are not present in this section.

Results

Provide more details in the description of Table 1, including the database from which the data were retrieved.

Specify the methodology used for generating each plot. For instance, state: "The following plot was generated using X package of Y software."

Discussion

In the discussion of the differential expression patterns of CCT genes in response to abiotic stress, include analysis related to the timing (e.g., hours after stress induction). Discuss how these time intervals relate to stress responses in plants.

Reviewer 2 Report

Comments and Suggestions for Authors

The work is aimed to investigate the CCT gene family role in photoperiodic flowering and environmental stress response in two Solanum species: tomato (Solanum lycopersicum L.) and eggplant (Solanum melongena L.)

A genome-wide characterization and expression profiling analysis of the CCT gene family was performed. These methods are consistent with the aims of the research.

Results represent a great reference point for future research about the CCT family role in Solanum species.

Generally, the work is well described and well organized.

However, some minor revisions could improve the quality of the work. More specifically:

                    In the “Introduction” section,

o   Actually, tomato and eggplant are not vegetables, I suggest to change/remove ‘vegetables’ with crops Lines 38 and 41.

o   Some recent references about the role of the CCT family role in Solanum species could be added, such as those available at

https://www.sciencedirect.com/science/article/abs/pii/S0141813024067151

                    In the “Results” section,

o   I suggest to include details about the used databases/tools in ‘Materials and Methods’ section, removing them from this section

o   The quality of some figures could be improved. In order to make them more readable, I suggest to split some figures in subfigures

                    In the ‘Discussion’ section,

o   The value added of the achieved results compared to the previous ones could be discussed

Some minor issues:

                    The abstract is too long. In fact, it must be a single paragraph of up to 200 words. Please check the template for this journal. Please focus on the main achieved results

                    In the references, many scientific crop names are not in Italic form.
